# Interaction of the Cosmic Dark Fluid with Dynamic Aether: Parametric Mechanism of Axion Generation in the Early Universe

**Alexander Balakin \***[ID]**, Alexei Ilin**[ID] **and Amir Shakirzyanov**[ID]

Department of General Relativity and Gravitation, Institute of Physics, Kazan Federal University, Kremlevskaya Str. 16a, Kazan 420008, Russia; alexeyilinjukeu@gmail.com (A.I.); shamirf@mail.ru (A.S.)
\* Correspondence: alexander.balakin@kpfu.ru

**Abstract:** We consider an isotropic homogeneous cosmological model with five interacting elements: first, the dynamic aether presented by a unit timelike vector field; second, the pseudoscalar field describing an axionic component of the dark matter; third, the cosmic dark energy, described by a rheologic fluid; fourth, the non-axionic dark matter coupled to the dark energy; and fifth, the gravity field. We show that the early evolution of the Universe described by this model can include two specific epochs: the first one can be characterized as a super-inflation epoch; the second epoch is associated with an oscillatory regime. The dynamic aether carries out a regulatory mission; the rheologic dark fluid provides the specific features of the spacetime evolution. The oscillations of the scale factor and of the Hubble function are shown to switch on the parametric (Floquet-type) mechanism of the axion number growth.

**Keywords:** alternative theories of gravity; axion dark matter; dynamic aether; rheologic dark fluid

**PACS:** 04.20.-q; 04.40.-b; 04.40.Nr; 04.50.Kd

## 1. Introduction

The paper is dedicated to the memory of Steven Weinberg (AB: half a century ago, the excellent book [1] predetermined my scientific life).

### 1.1. Motivation of the Work

#### 1.1.1. Inflation vs. Super-Inflation

We live and work in a unique situation, when we obtain new sensational results of observations made on the James Webb Space Telescope (JWST) practically in real time (for information see, e.g., the official website webb.nasa.gov (accessed on 23 August 2023)). These new data concern, in particular, the discovery of Mothra, an extremely magnified monster star; estimations of the masses of warm dark matter particles and of axion dark matter particles [2]; the abundance of carbon-containing molecules [3], etc. In addition, starting from the discovery of gravitational waves in 2015 [4], when two black holes with masses 36 $M_{\text{Sun}}$ and 29 $M_{\text{Sun}}$ collided, more than a hundred events of this type have been recorded. These events are associated with the merger and collision of black holes, whose masses are in the range from 20 to 90 solar masses.

Many of the data obtained by JWST and the LIGO–VIRGO collaboration are so unexpected that they make us think about the revision of the models of the early evolution of our Universe. What is, from our point of view, the main element of such a revision? We think that the starting period of the Universe evolution, when the size of the Universe does not exceed the size of the region allowing macroscopic causal processes, has to be longer than we think (of course, for comparison, we have to use the time scale that is predetermined by the corresponding value of the effective Hubble function $H$ associated with that epoch).

Such a possibility appears, for instance, if we consider super-inflation. The term super-inflation has been already used, e.g., in the model of Loop Quantum Cosmology [5], in order to mark a specific episode of the Universe inflationary evolution, when the kinetic energy of the scalar field is much more than the potential energy. Our goal is to consider not approximate but exact solutions describing the super-inflation as an alternative to standard inflation. Of course, the inflation scenario has already explained many details of the early Universe evolution, and when we pose a question about a super-inflation, we have to motivate this step. Keeping in mind this simple argument, we would like to attract attention to one new fact only. The LIGO–VIRGO collaboration has proved that black holes with intermediate masses (from 20 to 90 solar masses) do exist. Astrophysicists are ready to theoretically explain the presence of black holes with masses of several solar masses obtained in the scenario of a star collapse; also, one can explain the existence of super-massive black holes. However, now there is no adequate theory for the formation of the medium-sized black holes and super-massive stars. But if the causal period of the early Universe evolution lasted longer than the inflation theory predicts, we have a natural opportunity to explain the observed set of the black hole masses. We hope that a corresponding model for the formation of medium-sized black holes will be formulated in the near future: for example, similar to how the problem of the causal limit of the neutron star maximum mass was solved in [6].

Mathematically, one can explain the mentioned idea by comparing two functions $y_1 = e^{H(t-t_0)}$ and $y_2 = \exp[\alpha \sinh H(t-t_0)]$. Both functions start with the same values $y_1(t_0) = 1 = y_2(t_0)$. The first function describes the standard inflation, and we can require that at the moment $t = t_*$, the scale factor increased $10^{26}$ times, i.e., $\frac{a(t_*)}{a(t_0)} = e^{H(t_*-t_0)} = 10^{26}$. We obtain in this case the well-known 60 e-folds as follows: $H(t_*-t_0) = 26 \ln 10 \approx 60$. If we use the second function, which describes the so-called super-inflation, and again require that $\frac{a(t_*)}{a(t_0)} = \exp[\alpha \sinh H(t_*-t_0)] = 10^{26}$, we have to assume that $\alpha \approx 120 \cdot 10^{-26}$. Thus, for small values of time $H(t-t_0) \to 0$, we can decompose the mentioned functions as $y_1 \approx 1 + H(t-t_0)$ and $y_2 \approx 1 + 120 \cdot 10^{-26} H(t-t_0)$. Finally, we compare the time moments $t_1$ and $t_2$ for which the sizes of the expanding Universe become of the same order as the causal domain size $L_{\text{causal}}$. We obtain that $t_1-t_0 = 120 \cdot 10^{-26}(t_2-t_0)$, or equivalently, $t_2-t_0 \propto 10^{24}(t_1-t_0)$. This means that the super-exponential growth, on the one hand, guarantees that at $t = t_*$ the Universe expanded $10^{26}$ times as due to the standard inflation; on the other hand, the causal period of the Universe evolution lasts much longer, ensuring the development of causal phenomena, which could be associated, for example, with the formation of proto-galaxies, proto-stars, medium-sized black holes, etc.

Super-inflation can be described, in particular, as the exact solution of the set of master equations in the framework of models with a dark fluid of rheological type. In the works [7,8], we obtained the super-inflationary solution assuming that the equation of state of dark energy contains the convective derivative of the pressure; we also assumed that the dynamics of the dark matter particles is under control of the Archimedean-type force induced by dark energy. In the work [9], the super-inflationary solution appears as the exact solution of the dark fluid model with the kernel of contact interaction of the integral Volterra type. In the work [10], the mentioned solution appeared when we used the integral representation of the equations of state of the dark energy and dark matter. In other words, the solutions of the super-inflationary type seem to be typical ones for the rheological models of the cosmic dark fluid.

In this work, we consider again the rheological models for the dark fluid; however, now, we assume that the dark matter is a multi-component substratum [11]. We separate the axionic component of the dark matter [12] and consider axions on the language of field theory as a pseudoscalar field with modified periodic potential. Other components of the dark matter (WIMPs, ALPs, warm and hot dark matter parts, etc.) are unified and described as a dark medium with rheological properties.

### 1.1.2. Dynamic Aether as a Guiding Element of the Cosmic Evolution

The concept of the dynamic aether [13–16] gives the theorist a unique tool for modeling the process of controlling cosmic expansion. The dynamic aether is described by the unit-time-like global vector field, which is associated with the aether velocity four-vector $U^j$ and realizes the idea of a privileged frame of reference [17,18].

In addition to the geometric aspects of control, the dynamic aether can control the rhythm of life in the Universe. The fact is that the scalar of expansion defined as the divergence of the velocity four-vector $\Theta = \nabla_k U^k$ coincides with the tripled Hubble function $\Theta = 3H$, when the Universe is isotropic and homogeneous. Thus, for the spacetimes of the FLRW type, the scalar $\mathcal{T} = \frac{3}{\Theta}$ predetermines the typical time scale of the Universe evolution. Traditional history of the Universe is written using energy units and the equivalent temperature: the main stages of the Universe evolution are tied to some milestones associated with breaking of symmetry of fundamental interactions. However, for many purposes, we need to link the energy scale (or temperature scale) with the appropriate time scale. Clearly, requirements of the covariant approach do not give us possibility to use time-dependent parameters of equations of state or the time-dependent cosmological constant directly. Of course, one needs to introduce appropriate scalars, associated with some field, to solve the dynamic equation for this field and then to reconstruct the required guiding scalar. In this sense, the unit vector field $U^j$ is well suited for this role, since the evolution of this field is well described by the Jacobson equations [13–15], and the basic scalar $\Theta = \nabla_j U^j$ can be associated with the cosmological time scale.

Keeping in mind this idea we introduced in [19], the mechanism of the aetheric control on the axion field evolution is through introduction of the guiding function $\Phi_*(\Theta)$ into the axion field potential $V(\phi, \Phi_*(\Theta))$. Such a modification of the periodic axion potential led to the emergence of a concept of equilibrium states in the axion containing systems, for which the axion field $\phi$ takes the values $\phi = n\Phi_*$ with an integer $n$ [20]. In this context, the value of the expansion scalar $\Theta$ predetermines the position and depth of minima of the axion field potential. The extension of this approach has shown that for cosmological and astrophysical applications, it would be interesting to enlarge the number of guiding functions, which could be constructed using the covariant derivative of the aether velocity four-vector. For instance, when we deal with cosmological models of the Bianchi types, we cannot ignore the fact that in addition to the expansion scalar, we can use the non-vanishing symmetric traceless shear tensor describing the aether flow, $\sigma_{mn}$, in the theory. Correspondingly, the new geometric aspects of the Einstein–Maxwell–aether theory can be associated with the term $\sigma^{mn} F_{mp} F_n{}^p$ in the modified Lagrangian; the new geometric aspect of the Einstein–aether–axion theory can be connected with additional part of the axion kinetic energy $\sigma^{mn} \nabla_m \phi \nabla_n \phi$. As for the description of the aetheric control over the axion system, one can add the square of the shear tensor $\sigma^2 = \sigma_{mn} \sigma^{mn}$ as an argument of the guiding function $\Phi_*(\Theta, \sigma^2)$. For the static spherically symmetric model, the square of the acceleration four-vector $a^2$ may be in demand; the square of the antisymmetric vorticity tensor $\omega^2 = \omega_{mn} \omega^{mn}$ could appear as an argument of the guiding function in the model of Gödel type, describing the rotating Universe.

### 1.1.3. Interaction of the Dynamic Aether with Dark Fluid

All the mentioned extensions of the Einstein–aether theory are formulated in the language of field theory. The description of the dark fluid is made in terms of relativistic phenomenological hydrodynamics of two-component fluids. In order to realize the idea of aetheric control over the dark fluid, we suggest to include the scalars $\Theta$, $\sigma^2$, $a^2$, and $\omega^2$ into the Lagrangian $L_{(DF)}$ of the dark fluid. Such an approach can be indicated as the semi-phenomenological one. In this work, we restrict ourselves by the ansatz that the function $L_{(DF)}$ has a multi-step structure. This representation is based on the Heaviside step functions, arguments of which contain the expansion scalar $\Theta$. Using this approach, we assume that the aether divides the history of the Universe evolution into episodes, which can be indicated as inflation, super-inflation, oscillatory stage, etc. In this division into episodes, some critical values of the expansion scalar appear, $\Theta_*^{(1)}$, $\Theta_*^{(2)}$, $\Theta_*^{(3)}$, etc., which are assumed to play roles analogous to the roles of

critical temperatures $T_*^{(1)}$, $T_*^{(2)}$, and $T_*^{(3)}$ in the series of phase transitions associated with the Universe restructuring.

### 1.1.4. The Role of Axionic Dark Matter in Our Approach

We support the point of view that the axionic component is the key constructive element of the multi-component dark matter. The history of investigations (theoretical and experimental) of the axionic dark matter phenomenon (see, e.g., [21–29]) hints to us that the anomalous growth of the axion number in the early Universe could take place (the so-called "axionization" of the Universe); now, these relic axions form the basic part of cold dark matter. Following this idea, we consider two mechanisms of instability in the axion system provoked by the dark fluid controlled by the dynamic aether. The first mechanism can be realized in the scheme of super-inflationary expansion of the Universe. The second mechanism can be switched on at the oscillatory stage of the Universe evolution; it can be associated with the parametric instability described by the Floquet theorem for the Hill equation with periodic coefficients.

### 1.2. The Structure of the Work

The paper is organized as follows. In Section 2, we describe the mathematical formalism and derive the master equations of the presented theory. In Section 3, we consider applications of this theory to the model of evolution of the isotropic homogeneous Universe filled with the dynamic aether, two-component dark fluid, and axion field. In Section 3.1, we reduce the basic master equations to the chosen spacetime symmetry and find the exact solution to the equations for the unit vector field (aether velocity). In Section 3.2, we focus on the super-inflationary scenario of the Universe evolution; we find exact solutions for the scale factor and Hubble function, and we reconstruct the state functions for the rheologically active dark energy and non-axionic dark matter; in Section 3.2.3, we consider the problem of instability in the axion system controlled by the dynamic aether. In Section 3.3, we study the oscillatory episode of the Universe evolution; we again find the exact solution for the geometric quantities and for the dark fluid state functions; in Section 3.3.3, we focus on the analysis of the parametric mechanism of the axion generation. Section 4 includes discussion and conclusions.

## 2. The Formalism

### 2.1. The Total Action Functional

We consider the extension of the Einstein–aether–axion model and add the term associated with the dark fluid $L_{(\mathrm{DF})}$ to the total Lagrangian. The total action functional

$$-S_{(\mathrm{EA})} = \int d^4 x \sqrt{-g} \left\{ \frac{1}{2\kappa} \left[ R + 2\Lambda + \lambda(g_{mn}U^m U^n - 1) + K^{ab}_{\ \ mn} \nabla_a U^m \nabla_b U^n \right] + \frac{1}{2}\Psi_0^2 \left( V - \nabla_k \phi \nabla^k \phi \right) + L_{(\mathrm{DF})} \right\} \quad (1)$$

contains the standard Einstein–Hilbert term with the determinant of the metric $g$, the covariant derivative $\nabla_k$, the Ricci scalar $R$, the cosmological constant $\Lambda$, and the Einstein constant $\kappa = \frac{8\pi G}{c^4}$. The unit-time-like vector field $U^j$ is associated with the velocity four-vector of the dynamic aether (see, e.g., [13–16] for history, mathematical details, and definitions); the term with the Lagrange multiplier $\lambda$ in front is introduced to provide the normalization of the vector field, $U^k U_k = 1$. The constitutive tensor

$$K^{ab}_{\ \ mn} = C_1 g^{ab} g_{mn} + C_2 \delta^a_m \delta^b_n + C_3 \delta^a_n \delta^b_m + C_4 U^a U^b g_{mn} \quad (2)$$

contains four phenomenological constants $C_1$, $C_2$, $C_3$, and $C_4$. The pseudoscalar field $\phi$ stands to describe the axionic part of the dark matter; the term $V$ describes the potential of the axion field; the parameter $\Psi_0$ relates to the coupling constant of the axion–photon interaction $g_{A\gamma\gamma}$, $\frac{1}{\Psi_0} = g_{A\gamma\gamma}$. The potential of the axion field

$$V(\phi, \Phi_*) = \frac{m_A^2 \Phi_*^2}{2\pi^2} \left[ 1 - \cos\left( \frac{2\pi\phi}{\Phi_*} \right) \right] \quad (3)$$

is periodic since it inherits the discrete symmetry $\frac{2\pi\phi}{\Phi_*} \to \frac{2\pi\phi}{\Phi_*} + 2\pi n$ ($n$ is an integer). Also, this potential can be indicated as modified periodic potential since it contains the guiding function $\Phi_*$, which describes an averaged value of the pseudoscalar field; generally, this guiding function depends on coordinates via the scalars associated with the model as a whole (see, e.g., [19]). This periodic potential has its minima at $\phi = n\Phi_*$; for these values of the axion field, the potential and its first derivative vanish, $V_{|\phi=n\Phi_*} = 0$, $\frac{\partial V}{\partial\phi}_{|\phi=n\Phi_*} = 0$. We indicate the states $\phi = n\Phi_*$ as the equilibrium state. Near the minima, when $\phi \to n\Phi_* + \psi$ and $\left|\frac{2\pi\psi}{\Phi_*}\right|$ is small, the potential takes the standard form $V \to m_A^2 \psi^2$, where $m_A$ is the axion rest mass. The integer $n$ describes the level on which the axion field is fixed; when $n = 1$, we deal with the basic level; when $n = 0$, we assume that the axions are absent.

If the guiding function $\Phi_*$ is constant, the more convenient term can be used for this quantity, namely, the vacuum expectation value. In this regard, it is important to mention that, unlike the axion theory, the standard theory of the dynamic aether does not contain an appropriate vector field potential. In this sense, the vacuum expectation value of the vector field does not appear in the standard version of this theory; thus, there is no fixed direction in the space, and the spatial isotropy violation can not take place.

The term $L_{(DF)}$ describes the dark fluid, which indicates two interacting cosmic substrates: the dark energy and the non-axionic dark matter.

### 2.2. Auxiliary Elements of Analysis

Based on the velocity four-vector of the aether $U^j$, we decompose all the tensor quantities highlighting the longitudinal and transversal components. In particular, the covariant derivative can be decomposed as follows

$$\nabla_k = U_k D + \overset{\perp}{\nabla}_k, \quad D = U^s \nabla_s, \quad \overset{\perp}{\nabla}_k = \Delta_k^j \nabla_j, \quad \Delta_k^j = \delta_k^j - U^j U_k. \tag{4}$$

$\Delta_k^j$ is the projector. The covariant derivative $\nabla_k U_j$ can be decomposed in the standard sum

$$\nabla_k U_j = U_k DU_j + \sigma_{kj} + \omega_{kj} + \frac{1}{3}\Delta_{kj}\Theta, \tag{5}$$

where the acceleration four-vector $DU_j$, the symmetric traceless shear tensor $\sigma_{kj}$, the skew-symmetric vorticity tensor $\omega_{kj}$ and the expansion scalar $\Theta$ are presented by the well-known formulas

$$DU_j = U^s \nabla_s U_j, \quad \sigma_{kj} = \frac{1}{2}\left(\overset{\perp}{\nabla}_k U_j + \overset{\perp}{\nabla}_j U_k\right) - \frac{1}{3}\Delta_{kj}\Theta, \quad \omega_{kj} = \frac{1}{2}\left(\overset{\perp}{\nabla}_k U_j - \overset{\perp}{\nabla}_j U_k\right), \quad \Theta = \nabla_k U^k. \tag{6}$$

Using the decomposition (5) we form one linear and three quadratic scalars

$$\Theta = \nabla_k U^k, \quad a^2 = DU_k DU^k, \quad \sigma^2 = \sigma_{mn}\sigma^{mn}, \quad \omega^2 = \omega_{mn}\omega^{mn}. \tag{7}$$

Generally, the guiding function $\Phi_*$ is assumed to be the function of all four scalars, $\Phi_*(\Theta, a^2, \sigma^2, \omega^2)$, and we prepare the basic formulas for the general case. But below, we consider the application of the theory to the spatially isotropic homogeneous model, for which $DU^j = 0$, $\sigma_{mn} = 0$, $\omega_{mn} = 0$; naturally, we will focus on the case when the guiding function depends on the expansion scalar $\Theta$ only, i.e., $\Phi_* = \Phi_*(\Theta)$. The scalar $\mathcal{K} \equiv K^{abmn}(\nabla_a U_m)(\nabla_b U_n)$ can also be expressed via the mentioned invariants as follows:

$$\mathcal{K} = (C_1 + C_4)DU_k DU^k + (C_1 + C_3)\sigma_{ik}\sigma^{ik} + (C_1 - C_3)\omega_{ik}\omega^{ik} + \frac{1}{3}(C_1 + 3C_2 + C_3)\Theta^2. \tag{8}$$

Clearly, for the spatially isotropic and homogeneous model, this term reduces to $\mathcal{K} = \frac{1}{3}(C_1 + 3C_2 + C_3)\Theta^2$.

### 2.3. Master Equations of the Model

#### 2.3.1. Master Equations for the Unit Vector Field

Variations of the action functional (1) with respect to the Lagrange multiplier $\lambda$ and to the four-vector $U^i$ give the following set of equations, respectively:

$$g_{mn}U^m U^n = 1,\tag{9}$$

$$\nabla_a \mathcal{J}_j^a = \lambda U_j + C_4 DU_m \nabla_j U^m + \frac{1}{2}\kappa\Psi_0^2 \frac{\delta V}{\delta U^j} + \kappa\frac{\delta L_{(\mathrm{DF})}}{\delta U^j}.\tag{10}$$

The tensor $\mathcal{J}_j^a$ is of the form

$$\mathcal{J}_j^a = K^{ab}_{jn}\nabla_b U^n = C_1 \nabla^a U_j + C_2 \delta_j^a \Theta + C_3 \nabla_j U^a + C_4 U^a DU_j.\tag{11}$$

The variational derivatives in (10) can be represented as follows:

$$\frac{1}{2}\kappa\Psi_0^2 \frac{\delta V}{\delta U^j} \Rightarrow -\nabla_j\left(\Omega\frac{\partial\Phi_*}{\partial\Theta}\right) - 2DU_j D\left(\Omega\frac{\partial\Phi_*}{\partial a^2}\right) - \nabla^n\left(2\Omega\frac{\partial\Phi_*}{\partial\sigma^2}\sigma_{jn}\right) + \nabla^n\left(2\Omega\frac{\partial\Phi_*}{\partial\omega^2}\omega_{jn}\right) +$$
$$+2\Omega\frac{\partial\Phi_*}{\partial a^2}\left[DU_k\nabla_j U^k - \Theta DU_j - D^2 U_j\right] - 2\Omega\frac{\partial\Phi_*}{\partial\sigma^2}DU^n \sigma_{jn} - 2\Omega\frac{\partial\Phi_*}{\partial\omega^2}DU^n \omega_{jn},\tag{12}$$

$$\kappa\frac{\delta L_{(\mathrm{DF})}}{\delta U^j} \Rightarrow \kappa\left\{-\nabla_j\left(\frac{\partial L_{(\mathrm{DF})}}{\partial\Theta}\right) - 2DU_j D\left(\frac{\partial L_{(\mathrm{DF})}}{\partial a^2}\right) - \nabla^n\left(2\frac{\partial L_{(\mathrm{DF})}}{\partial\sigma^2}\sigma_{jn}\right) + \nabla^n\left(2\frac{\partial L_{(\mathrm{DF})}}{\partial\omega^2}\omega_{jn}\right) +\right.$$
$$\left.+2\frac{\partial L_{(\mathrm{DF})}}{\partial a^2}\left[DU_k\nabla_j U^k - \Theta DU_j - D^2 U_j\right] - 2\frac{\partial L_{(\mathrm{DF})}}{\partial\sigma^2}DU^n \sigma_{jn} - 2\frac{\partial L_{(\mathrm{DF})}}{\partial\omega^2}DU^n \omega_{jn}\right\}.\tag{13}$$

Here, we used the convenient definition

$$\Omega = \frac{1}{2}\kappa\Psi_0^2 \frac{\partial V}{\partial\Phi_*} = \frac{\kappa\Psi_0^2 m_A^2}{2\pi^2}\left\{\Phi_*\left[1 - \cos\left(\frac{2\pi\phi}{\Phi_*}\right)\right] - \pi\phi\sin\left(\frac{2\pi\phi}{\Phi_*}\right)\right\}.\tag{14}$$

The Lagrange multiplier can be formally presented as the sum $\lambda = \lambda_{(\mathrm{U})} + \lambda_{(\mathrm{V})} + \lambda_{(\mathrm{DF})}$, where

$$\lambda_{(\mathrm{U})} = U^j \nabla_a \mathcal{J}_j^a - C_4 DU_m DU^m, \quad \lambda_{(\mathrm{V})} = -\frac{1}{2}\kappa\Psi_0^2 U^j \frac{\delta V}{\delta U^j}, \quad \lambda_{(\mathrm{DF})} = -\kappa U^j \frac{\delta L_{(\mathrm{DF})}}{\delta U^j}.\tag{15}$$

Using (12) and (13), we obtain immediately

$$\lambda_{(\mathrm{V})} = D\left(\Omega\frac{\partial\Phi_*}{\partial\Theta}\right) - 2\Omega\sigma^2\frac{\partial\Phi_*}{\partial\sigma^2} - 2\Omega\omega^2\frac{\partial\Phi_*}{\partial\omega^2} - 4\Omega a^2\frac{\partial\Phi_*}{\partial a^2},\tag{16}$$

$$\lambda_{(\mathrm{DF})} = \kappa\left\{D\left(\frac{\partial L_{(\mathrm{DF})}}{\partial\Theta}\right) - 2\sigma^2\frac{\partial L_{(\mathrm{DF})}}{\partial\sigma^2} - 2\omega^2\frac{\partial L_{(\mathrm{DF})}}{\partial\omega^2} - 4a^2\frac{\partial L_{(\mathrm{DF})}}{\partial a^2}\right\}.\tag{17}$$

When we deal with the spatially isotropic homogeneous model, only the first terms in the right-hand sides of the Equations (16) and (17) are non-vanishing. In the state of the axionic equilibrium, when $\phi = n\Phi_*$, we obtain that $\Omega = 0$.

#### 2.3.2. Master Equation for the Axion Field

Variation of the total action functional with respect to the axion field yields

$$\nabla^k\nabla_k\phi + \frac{m_A^2\Phi_*}{2\pi}\sin\left(\frac{2\pi\phi}{\Phi_*}\right) = 0.\tag{18}$$

If we deal with the basic equilibrium state of the axion field and put $\phi = \Phi_*$ into (18), we can conclude that the guiding function $\Phi_*$ satisfies the massless Klein–Gordon equation

$$\nabla^k \nabla_k \Phi_* = 0 \,. \tag{19}$$

2.3.3. Master Equations for the Gravitational Field

Variation with respect to the metric gives the gravity field equations:

$$R_{ik} - \frac{1}{2} R g_{ik} - \Lambda g_{ik} = T_{ik}^{(\text{U})} + \kappa T_{ik}^{(\text{A})} + \kappa T_{ik}^{(\text{DF})} \,. \tag{20}$$

The first term in the right-hand side of (20) is the standard stress–energy tensor associated with the aether flow [13–15]:

$$T_{ik}^{(\text{U})} = \frac{1}{2} g_{ik} \, K^{abmn} \nabla_a U_m \nabla_b U_n + \lambda_{(\text{U})} U_i U_k + \tag{21}$$

$$+ \nabla^m \left[ U_{(i} \mathcal{J}_{k)m} - \mathcal{J}_{m(i} U_{k)} - \mathcal{J}_{(ik)} U_m \right] +$$

$$+ C_1 [(\nabla_m U_i)(\nabla^m U_k) - (\nabla_i U_m)(\nabla_k U^m)] + C_4 D U_i D U_k \,.$$

The parentheses symbolize the symmetrization of indices. The term $\lambda_{(\text{U})} U_i U_k$ is included into $T_{ik}^{(\text{U})}$ since $\lambda_{(\text{U})}$ (see (15)) contains the aether velocity and its derivatives only. The second term

$$\kappa T_{ik}^{(\text{A})} = \kappa \Psi_0^2 \left[ \nabla_i \phi \nabla_k \phi + \frac{1}{2} g_{ik} (V - \nabla_s \phi \nabla^s \phi) \right] + \lambda_{(\text{V})} U_i U_k - g_{ik} (D + \Theta) \left( \Omega \frac{\partial \Phi_*}{\partial \Theta} \right) + \tag{22}$$

$$+ 2 \Omega \frac{\partial \Phi_*}{\partial a^2} D U_i D U_k + 2 \nabla_s \left\{ \Omega \frac{\partial \Phi_*}{\partial a^2} \left[ D U^s U_i U_k - 2 U^s U_{(i} D U_{k)} \right] \right\} -$$

$$- 2 \nabla_s \left[ \Omega \frac{\partial \Phi_*}{\partial \sigma^2} U^s \sigma_{ik} \right] - 4 \Omega \frac{\partial \Phi_*}{\partial \sigma^2} \left[ D U_n \sigma_{(i}^n U_{k)} + \sigma_{(i}^n \omega_{k)n} \right] +$$

$$+ 4 \nabla_s \left[ \Omega \frac{\partial \Phi_*}{\partial \omega^2} U_{(i} \omega_{k).}^s \right] - 4 \Omega \frac{\partial \Phi_*}{\partial \omega^2} \left[ D U^n U_{(i} \omega_{k)n} + \sigma_{(i}^n \omega_{k)n} \right]$$

describes the contribution of the axion field coupled to the aether. The term $\lambda_{(\text{V})} U_i U_k$ is included into this part of the total stress–energy tensor, since it describes the interaction of the aether with the axion field. The stress–energy tensor of the dark fluid $T_{ik}^{(\text{DF})}$ is presented by two terms:

$$T_{ik}^{(\text{DF})} = \frac{1}{\kappa} \lambda_{(\text{DF})} U_i U_k + \frac{(-2)}{\sqrt{-g}} \frac{\delta}{\delta g^{ik}} \left[ \sqrt{-g} L_{(\text{DF})} \right] \,. \tag{23}$$

The first term $\frac{1}{\kappa} \lambda_{(\text{DF})} U_i U_k$ (see (15)) is the last element of the total term $\lambda U_i U_k$, and the second one is given by the standard variation formula. This last construction can be decomposed algebraically using the aether velocity four-vector $U^j$,s and we obtain

$$T_{ik}^{(\text{DF})} = \left[ \frac{1}{\kappa} \lambda_{(\text{DF})} + \mathcal{W} \right] U_i U_k + q_i U_k + q_k U_i + \mathcal{P}_{ik} \,. \tag{24}$$

Here, $\mathcal{W}$ denotes the total energy density of the dark fluid; $q_j$ is the heat-flux four-vector, and $\mathcal{P}_{ik}$ is the pressure tensor; these quantities are defined in the standard way:

$$\frac{1}{\kappa} \lambda_{(\text{DF})} + \mathcal{W} = U^i T_{ik}^{(\text{DF})} U^k \,, \quad q_j = \Delta_j^i T_{ik}^{(\text{DF})} U^k \,, \quad \mathcal{P}_{ik} = \Delta_i^p T_{pq}^{(\text{DF})} \Delta_k^q \,. \tag{25}$$

When we deal with the isotropic homogeneous cosmological model, we have to put $q_j = 0$ and $\mathcal{P}_{ik} = -\mathcal{P} \Delta_{ik}$. We assume that $\mathcal{W} = W + E$ is the sum of the energy densities of

the dark energy and of the non-axionic dark matter; similarly, $\mathcal{P} = P + \Pi$ is the sum of the corresponding scalars of pressure.

### 2.3.4. Bianchi Identity and Conservation Law

The Bianchi identity requires that

$$
\nabla^k \left[ T_{ik}^{(\mathrm{U})} + \kappa T_{ik}^{(\mathrm{A})} + \kappa T_{ik}^{(\mathrm{DF})} \right] = 0 \,. \tag{26}
$$

We would like to emphasize two important details. First, since we included the term $\lambda_{(\mathrm{U})} U_i U_k$ into the stress–energy tensor $T_{ik}^{(\mathrm{U})}$, we have guaranteed that $\nabla^k T_{ik}^{(\mathrm{U})} = 0$ on the solutions of the master Equation (10). Second, the presence of the term $\lambda_{(\mathrm{V})} U_i U_k$ in the stress–energy tensor $T_{ik}^{(\mathrm{A})}$ guarantees that $\nabla^k T_{ik}^{(\mathrm{A})} = 0$ on the solutions of the master Equation (18). Thus, the equality $\nabla^k T_{ik}^{(\mathrm{DF})} = 0$ is the consequence of the Bianchi identity on the one hand, and it gives the evolutionary equation for the state functions of the dark fluid on the other hand.

## 3. Application to the Spatially Isotropic Homogeneous Cosmological Model

### 3.1. The Spacetime Platform and Reduced Master Equations

We work below with the spacetime of the FLRW type with the metric

$$
ds^2 = dt^2 - a^2(t) \left( dx^2 + dy^2 + dz^2 \right) . \tag{27}
$$

The symmetry of the model hints that one can search for the velocity four-vector of the aether in the form $U^j = \delta_0^j$. It is well known that for this metric, the covariant derivative of the vector field

$$
\nabla_k U_i = \frac{1}{2} \dot{g}_{ik} \tag{28}
$$

is symmetric, and we see explicitly that $DU_j = 0$, $\sigma_{mn} = 0$, $\omega_{mn} = 0$. The expansion scalar is non-vanishing

$$
\Theta = 3\frac{\dot{a}}{a} = 3H(t) \Rightarrow \nabla_k U_i = \Delta_{ik} H(t) \,. \tag{29}
$$

Here, $H(t) = \frac{\dot{a}}{a}$ is the Hubble function, and the dot denotes, as usual, the derivative with respect to the cosmological time $t$ (here and below $c = 1$).

### 3.1.1. Solution to the Equations of the Vector Field

When we start to analyze the reduced Equations (10) and (11), we have to emphasize the following circumstance. The sum of the coupling constants $C_1 + C_3$ can be estimated as $-6 \times 10^{-15} < C_1 + C_3 < 1.4 \times 10^{-15}$, and below, we put $C_1 + C_3 = 0$. This estimation is based on the result of observation of the binary neutron star merger (the events GW170817 and GRB 170817A [30]), which has shown that the ratio of the velocities of the gravitational and electromagnetic waves satisfies the inequalities $1 - 3 \times 10^{-15} < \frac{v_{\mathrm{gw}}}{c} < 1 + 7 \times 10^{-16}$), while according to [15], the square of the velocity of the tensorial aether mode is equal to $S_{(2)}^2 = \frac{1}{1-(C_1+C_3)}$. Keeping in mind that $DU_j = 0$, $\sigma_{mn} = 0$, and $\omega_{mn} = 0$, we find that (11) converts into

$$
J_j^a = C_2 \Theta \delta_j^a \,, \tag{30}
$$

and the equations for the unit vector field (10) now takes the form

$$
C_2 \nabla_j \Theta = \lambda U_j - \nabla_j \left( \Omega \frac{\partial \Phi_*}{\partial \Theta} \right) - \kappa \nabla_j \left( \frac{\partial L_{(\mathrm{DF})}}{\partial \Theta} \right) . \tag{31}
$$

Clearly, the set of Equation (31) contains only one non-trivial equation, which gives, in fact, the solution for the Lagrange multiplier $\lambda$:

$$D\left(C_2\Theta + \Omega\frac{\partial\Phi_*}{\partial\Theta} + \kappa\frac{\partial L_{(DF)}}{\partial\Theta}\right) = \lambda. \tag{32}$$

### 3.1.2. Reduced Equation for the Axion Field

For the fixed spacetime symmetry, the Equation (18) takes the form

$$\ddot{\phi} + 3H\dot{\phi} + \frac{m_A^2\Phi_*}{2\pi}\sin\left(\frac{2\pi\phi}{\Phi_*}\right) = 0. \tag{33}$$

Let us imagine that the axion field $\phi$ is frozen in the first minimum of the axion potential; it coincides with the guiding function, and thus, it follows the variations of $\Phi_*(t)$. Then, we put $\phi = \Phi_*$ into (33), and we have to admit that the axion field remains in the first minimum if the guiding function $\Phi_*$ satisfies the equation

$$\ddot{\Phi}_* + 3H\dot{\Phi}_* = 0. \tag{34}$$

Our ansatz is that (34) defines the missing master equation for the guiding function $\Phi_*$ in the framework of the established model.

Clearly, the Equation (34) admits the first integral

$$\dot{\Phi}_*(t) = \frac{\text{const}}{a^3(t)} = \dot{\Phi}_*(t_0)\left[\frac{a(t_0)}{a(t)}\right]^3. \tag{35}$$

Here and below, the parameter $t_0$ describes the time moment, starting from which our semi-phenomenological model is valid. We assume that $t_0 > t_{\text{quantum}}$, so that the results of evolution on the stage associated with the quantum description of the Universe are fixed in the initial data, e.g., in $\dot{\Phi}_*(t_0)$.

### 3.1.3. Key Equations for the Gravity Field

Solving the Einstein equations for the isotropic homogeneous cosmological model, we follow the standard scheme: we consider the gravity field Equation (20) for the indices $i = 0$, $k = 0$, and the consequence of the Bianchi identity (26) for $i = 0$, thus obtaining the first and second key equations of the model. Equation (18) for the axion field forms the third key equation. The first key equation can be written as

$$3H^2\left(1 + \frac{3}{2}C_2\right) - \Lambda = \kappa\left(\frac{\partial L_{(DF)}}{\partial\Theta}\right)^{\cdot} + \frac{1}{2}\kappa\Psi_0^2\left(V + \dot{\phi}^2\right) - \Omega\Theta\frac{\partial\Phi_*}{\partial\Theta} + \kappa(W + E). \tag{36}$$

The second key equation is presented by the balance equation for the dark fluid

$$\dot{W} + \dot{E} + 3H(W + E + P + \Pi) = -\kappa\left[\left(\frac{\partial L_{(DF)}}{\partial\Theta}\right)^{\cdot\cdot} + 3H\left(\frac{\partial L_{(DF)}}{\partial\Theta}\right)^{\cdot}\right]. \tag{37}$$

### 3.2. Super-Inflationary Scenario of Early Universe Evolution

The novelty of our approach is associated with a multi-step-like structure of the function $L_{(DF)}$. Let us explain this idea for the one-step function. We assume that

$$L_{(DF)} = \eta(\Theta_* - \Theta)L_{(DF)}^{(1)} + \eta(\Theta - \Theta_*)L_{(DF)}^{(2)}. \tag{38}$$

Here, $\eta(F)$ is the Heaviside function, which is equal to one if the argument $F$ is positive, $F > 0$, and is equal to zero when $F < 0$. The constant $\Theta_* \equiv \Theta(t_*)$ is the value of the expansion scalar at some fixed time moment $t_* > t_0$. We assume that both functions $L_{(DF)}^{(1)}$ and $L_{(DF)}^{(2)}$

depend on time, but do not include $\Theta$. Generally, $L_{(DF)}^{(1)}(t) \neq L_{(DF)}^{(2)}(t)$; however, the continuity condition is met, $L_{(DF)}^{(1)}(t_*) = L_{(DF)}^{(2)}(t_*)$. This means that we have to solve the key equations of the model for two intervals: $t_0 < t < t_*$ and $t > t_*$ and sew the found state functions at $t = t_*$. In fact, we guarantee that the derivative $\left(\frac{\partial L_{(DF)}}{\partial \Theta}\right)^{\cdot}$ vanishes in both mentioned intervals, as well as in the point $t = t_*$. To be more precise, we obtain

$$\frac{\partial L_{(DF)}}{\partial \Theta} = \delta(\Theta - \Theta_*)\left[L_{(DF)}^{(2)} - L_{(DF)}^{(1)}\right] = 0, \tag{39}$$

if the function $\Theta(t)$ is monotonic and the equation $\Theta(t_*) = \Theta_*$ has only one solution $t_*$.

3.2.1. Epoch of the Dark Fluid Domination

Let us assume that during the first episode of the Universe evolution, when $t_0 < t < t_*$, the axion field is absent, $\phi = 0$; thus, $\Omega = 0$ and $V = 0$, so the key equations of the model as a whole are reduced to the following two equations:

$$H^2 = \frac{\Lambda_*}{3} + \frac{\kappa_*}{3}(W + E), \tag{40}$$

$$\dot{W} + \dot{E} + 3H(W + E + P + \Pi) = 0. \tag{41}$$

Here, we used the definitions $\Lambda_* = \frac{\Lambda}{1 + \frac{3}{2}C_2}$ and $\kappa_* = \frac{\kappa}{1 + \frac{3}{2}C_2}$ and assumed that $C_2 > -\frac{2}{3}$. The next convenient step is to introduce the new variable $x = \frac{a(t)}{a(t_0)}$, providing the following relationships:

$$\dot{W} = xH\frac{dW}{dx}, \quad t - t_0 = \int_1^{\frac{a(t)}{a(t_0)}} \frac{dx}{xH(x)}. \tag{42}$$

In these terms, Equation (41) can be rewritten as

$$xW' + 3(W + P) = -\left[xE' + 3(E + \Pi)\right]. \tag{43}$$

Following the idea of modeling of the so-called kernel of interaction $Q$ (see, e.g., [31,32]), we can rewrite (43) as a pair of equations

$$xW' + 3(W + P) = Q(x), \quad xE' + 3(E + \Pi) = -Q(x). \tag{44}$$

The next step is to propose the constitutive equations for dark energy and for non-axionic dark matter and to formulate the structure of the kernel $Q(x)$. We assume that during the first episode of early Universe evolution, the following relationships are appropriate:

$$P = (\Gamma - 1)W, \quad \Pi = (\gamma - 1)E, \quad Q(x) = K_0 \int_1^x \frac{dy}{y}[E(y) - W(y)]. \tag{45}$$

Here, $\Gamma$, $\gamma$, and $K_0$ are some phenomenological constants. The representation of the kernel $Q$ in the integral form was motivated in the work [9]. Using (45), we obtain the integral equation of the Volterra type

$$xW' + 3\Gamma W = K_0 \int_1^x \frac{dy}{y}(E(y) - W(y)), \tag{46}$$

from which one can extract the energy density of the non-axionic dark matter

$$E(x) = W(x) + \frac{1}{K_0}\left[x^2 W'' + xW'(1 + 3\Gamma)\right]. \tag{47}$$

Then, we put the function $E(x)$ into (43) and obtain the Euler equation of the third order for the function $W(x)$:

$$x^3 W''' + 3x^2 W''(1 + \Gamma + \gamma) + x W'[2K_0 + (1 + 3\Gamma)(1 + 3\gamma)] + 3K_0(\Gamma + \gamma)W = 0. \quad (48)$$

The characteristic equation for this differential Euler equation is of the form

$$\sigma^3 + 3(\Gamma + \gamma)\sigma^2 + \sigma(2K_0 + 9\Gamma\gamma) + 3K_0(\Gamma + \gamma) = 0. \quad (49)$$

We are especially interested in the analysis of the case with the threefold degenerate root $\sigma_1 = \sigma_2 = \sigma_3 = 0$ of this characteristic equation. Clearly, such a situation takes place when

$$\Gamma + \gamma = 0, \quad K_0 = \frac{9}{2}\gamma^2. \quad (50)$$

For this special case, the solutions for $W(x)$ and $E(x)$ are of the form

$$W(x) = W(1)(1 - 3\Gamma \log x) + \frac{9}{4}\Gamma^2[W(1) + E(1)] \log^2 x, \quad (51)$$

$$E(x) = E(1)(1 + 3\Gamma \log x) + \frac{9}{4}\Gamma^2[W(1) + E(1)] \log^2 x. \quad (52)$$

Here, we took into account the consequence of the integral relationship (46), which gives $W'(1) = -3\Gamma W(1)$ (similarly, we see that $E'(1) = 3\Gamma E(1)$). For the square of the Hubble function, we obtain from (40)

$$\frac{3}{\kappa_*}\left[H^2(x) - H^2(1)\right] = 3\Gamma[E(1) - W(1)] \log x + \frac{9}{2}\gamma^2[W(1) + E(1)] \log^2 x, \quad (53)$$

where the following definition was used:

$$H^2(1) = \frac{\Lambda_*}{3} + \frac{\kappa_*}{3}[W(1) + E(1)]. \quad (54)$$

3.2.2. Exact Explicit Solutions in the Model of Super-Inflation

One can obtain now the analytic formula which links the scale factor and the cosmological time for arbitrary initial data $W(1)$ and $E(1)$. For illustration only, we put $E(1) = W(1)$ and obtain the integral

$$t - t_0 = \pm \int_1^{\frac{a(t)}{a(t_0)}} \frac{dx}{x\sqrt{H^2(t_0) + 3\kappa_*\gamma^2 W(t_0) \log^2 x}}. \quad (55)$$

We choose the sign plus in this formula, and direct integration gives the super-exponential law of evolution of the scale factor

$$\log\left[\frac{a(t)}{a(t_0)}\right] = \frac{H(t_0)}{\gamma\sqrt{3\kappa_* W(t_0)}} \sinh\left[\gamma\sqrt{3\kappa_* W(t_0)}(t - t_0)\right]. \quad (56)$$

When $W(t_0)$ tends to zero, i.e., dark fluid is absent, this formula gives the de Sitter-type law

$$\lim_{W(t_0) \to 0}\left[\frac{a(t)}{a(t_0)}\right] = e^{\sqrt{\frac{\Lambda_*}{3}}(t - t_0)} \quad (57)$$

with the aetherically modified cosmological constant $\Lambda_* = \frac{\Lambda}{1 + \frac{3}{2}C_2}$ (see Figure 1 for illustration).

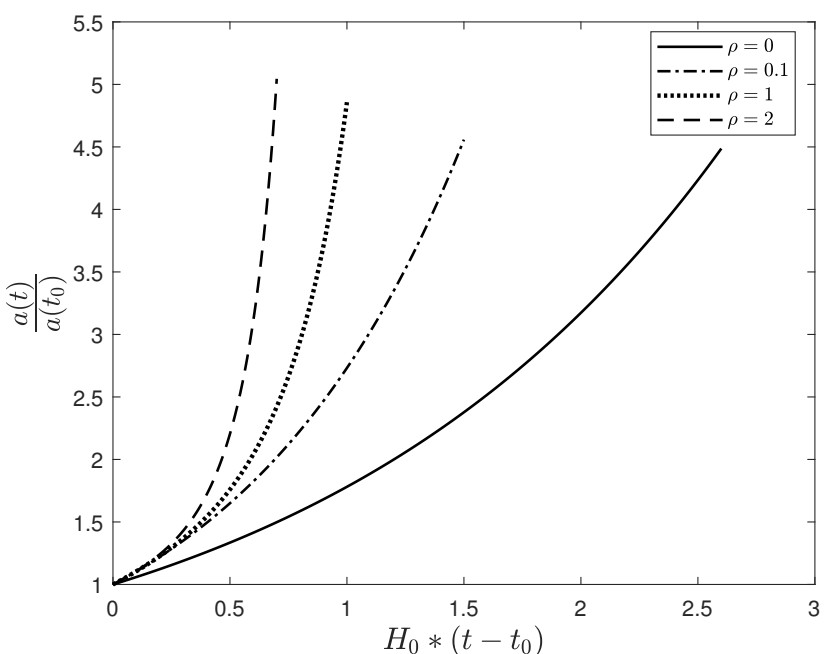

**Figure 1.** Illustration of the behavior of the super-exponential function $\frac{a(t)}{a(t_0)} = \exp\left\{\frac{1}{\rho}\sinh\rho H_0(t-t_0)\right\}$ for four values of the dimensionless parameter $\rho = \frac{1}{H_0}\gamma\sqrt{3\kappa_* W(t_0)}$. For illustration, we put $H(t_0) = H_0 = 1$. The curve with $\rho = 0$ corresponds to the de Sitter type regime (inflation).

In terms of the cosmological time, the Hubble function $H(t)$ is presented by the function

$$H(t) = H(t_0)\cosh\left[\gamma\sqrt{3\kappa_* W(t_0)}(t-t_0)\right], \tag{58}$$

which grows monotonically. The acceleration parameter is of the form

$$-q(t) \equiv \frac{\ddot{a}}{aH^2} = 1 + \frac{\gamma\sqrt{3\kappa_* W(t_0)}}{H(t_0)}\left\{\frac{\sinh\left[\gamma\sqrt{3\kappa_* W(t_0)}(t-t_0)\right]}{\cosh^2\left[\gamma\sqrt{3\kappa_* W(t_0)}(t-t_0)\right]}\right\}. \tag{59}$$

The energy density scalars for the dark energy and for the non-axionic dark matter take the form, respectively,

$$W(t) = W(t_0)\left\{1 + \frac{\sqrt{3}H(t_0)}{\sqrt{\kappa_* W(t_0)}}\sinh\left[\gamma\sqrt{3\kappa_* W(t_0)}(t-t_0)\right] + \frac{3H^2(t_0)}{2\kappa_* W(t_0)}\sinh^2\left[\gamma\sqrt{3\kappa_* W(t_0)}(t-t_0)\right]\right\}, \tag{60}$$

$$E(t) = W(t_0)\left\{1 - \frac{\sqrt{3}H(t_0)}{\sqrt{\kappa_* W(t_0)}}\sinh\left[\gamma\sqrt{3\kappa_* W(t_0)}(t-t_0)\right] + \frac{3H^2(t_0)}{2\kappa_* W(t_0)}\sinh^2\left[\gamma\sqrt{3\kappa_* W(t_0)}(t-t_0)\right]\right\}. \tag{61}$$

### 3.2.3. On the Stability of the Axion Field Configuration

We assumed above that the axion field is in the equilibrium state with $n = 0$. Since the scale factor is found, we can calculate guiding function $\Phi_*(t)$ using (35), obtaining the formal integral

$$\Phi_*(t) = \Phi_*(t_0) + \dot{\Phi}_*(t_0)\int_{t_0}^{t}d\tau\exp\left\{-\frac{\sqrt{3}H(t_0)}{\gamma\sqrt{\kappa_* W(t_0)}}\sinh\left[\gamma\sqrt{3\kappa_* W(t_0)}(\tau-t_0)\right]\right\}. \tag{62}$$

This integral cannot be presented via elementary functions, but can be expressed via incomplete modified Bessel functions [33]. Clearly, if we consider $\dot{\Phi}_*(t_0) = 0$ and $\Phi_*(t_0) = 2\pi$,

we obtain the potential $V(\phi) = 2m_A^2[1 - \cos\phi]$, which does not admit control over the axion evolution carried out by the dynamic aether.

Now, we admit that a fluctuation occurs $\phi \to \psi \neq 0$ with $\left|\frac{2\pi\psi}{\Phi_*}\right| \ll 1$, so that the master equation for the pseudoscalar field converts into

$$\ddot{\psi} + 3H(t)\dot{\psi} + m_A^2\psi = 0, \tag{63}$$

where $H(t)$ is given by (58). Using the replacement $\psi \to \left[\frac{a(t)}{a(t_0)}\right]^{-\frac{3}{2}}\Psi$, we obtain the so-called modified Hill equation

$$\ddot{\Psi} = \mathcal{J}(t)\Psi, \quad \mathcal{J}(t) = -m_A^2 + \frac{9}{4}H^2(t) + \frac{3}{2}\dot{H} = \tag{64}$$

$$= \left[\frac{9}{8}H^2(t_0) - m_A^2\right] + \frac{9}{8}H^2(t_0)\cosh\left[2\gamma\sqrt{3\kappa_*W(t_0)}(t-t_0)\right] +$$

$$+ \frac{3}{2}H(t_0)\gamma\sqrt{3\kappa_*W(t_0)}\sinh\left[\gamma\sqrt{3\kappa_*W(t_0)}(t-t_0)\right].$$

This terminology is based on the idea that if we replace the hyperbolic functions cosh and sinh by the trigonometric cos and sin, we obtain the standard Hill equation with periodic Hill potential. The Hill potential $\mathcal{J}(t)$ (64) is monotonic; thus, there exists a moment $t = t_+$, so that $\mathcal{J}(t > t_+) > 0$. In other words, when $t > t_+$, the Hill potential is positive, meaning that the term $\frac{\ddot{\Psi}}{\Psi}$ is also positive (see (64)). This means that if $\Psi(t_+) > 0$, the function $\Psi$ grows exponentially, while the factor $\left[\frac{a(t)}{a(t_0)}\right]^{-\frac{3}{2}}$ means that the function $\psi = \Psi\left[\frac{a(t)}{a(t_0)}\right]^{-\frac{3}{2}}$ decreases super-exponentially. Clearly, the function $\psi(t)$ grows; then, it reaches the maximum at $t = t_{\max}$ and then decreases inevitably under the influence of the Universe expansion. One can use the following estimation of the behavior of this perturbation:

$$\psi(t) \propto \cosh\left[2\gamma\sqrt{3\kappa_*W(t_0)}(t-t_0)\right]\exp\left\{-\frac{\sqrt{3}H(t_0)}{2\gamma\sqrt{\kappa_*W(t_0)}}\sinh\left[\gamma\sqrt{3\kappa_*W(t_0)}(t-t_0)\right]\right\}. \tag{65}$$

We deal with the instability in the axion field behavior on the interval $t_+ < t < t_{\max}$. Indeed, when $W(t_0) \neq 0$, the function $\psi(t)$ starts to grow and reaches the maximal value, with the height of the maximum determined by the initial value $W(t_0)$ (see Figure 2 for illustration).

The special case $W(t_0) = 0$ relates to the case of constant Hubble function $H > 0$, for which Equation (63) converts into the differential equation with constant coefficients, and the set of solutions of the corresponding characteristic equation

$$k_{1,2} = -\frac{3}{2}H \pm \sqrt{\frac{9}{4}H^2 - m^2} \tag{66}$$

do not contain real positive roots.

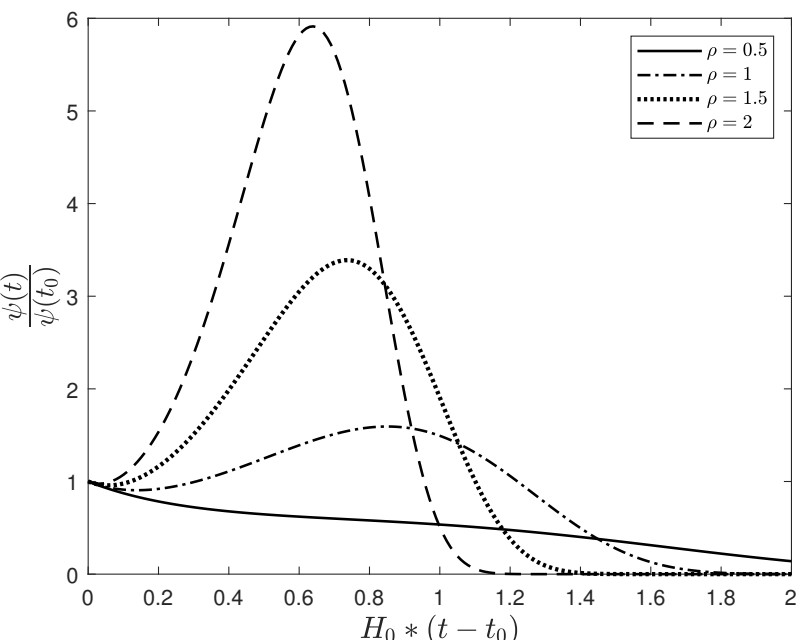

**Figure 2.** Illustration of the behavior of the axion field perturbation $\psi(t)$, which satisfies Equation (63), when the Hubble function describing the super-inflationary regime is given by (58).

### 3.3. Periodic Episode of the Early Universe Evolution

3.3.1. Modifications of the Model

Now, we assume that on the quantum stage of the Universe evolution, which corresponds to the time interval $0 < t < t_0$, standard inflation already led to the expansion characterized by 60 e-folds. We expect that the oscillatory regime of the Universe evolution will be switched on; however, in itself, such an event is unlikely. But, the aether, which controls the behavior of the dark fluid, is able to organize a fine-tuning to provide such an opportunity. Indeed, let us consider the following composite dark fluid model: first, during the interval $t_0 < t < t_*$, the dark fluid is described in the same way as in the previous section; but, at the moment $t = t_*$, the aether organizes the restructuring of the equations of state for the dark energy and non-axionic dark matter. Now, we deal with $\Gamma = 0$ for typical dark energy, and $\gamma = 1$ for typical dust. Also, we assume that the kernel of the contact interaction in the dark fluid is of the local type, but the dark energy now possesses self-interaction of the rheological type (see [10] for details). Mathematically, this means that we use the replacements

$$Q(x) = K_0 \int_{x_*}^{x} \frac{dy}{y} [E(y) - W(y)] \Rightarrow \tilde{Q}(x) = \omega_0 [E(x) - W(x)], \tag{67}$$

$$P(x) = (\Gamma - 1)W(x) \Rightarrow \tilde{P}(x) = -W + K_1 \int_{x_*}^{x} \frac{dy}{y} \left(\frac{y}{x}\right)^{\nu} W(y). \tag{68}$$

Here, $x_* = \frac{a(t_*)}{a(t_0)}$. Then, instead of (44) with (45), we consider the modified coupled balance equations

$$xW' + 3K_1 \int_{x_*}^{x} \frac{dy}{y} \left(\frac{y}{x}\right)^{\nu} W(y) = \omega_0(E - W), \quad xE' + 3E = \omega_0(W - E). \tag{69}$$

One can derive the key differential equations of the third order for $E(x)$ from the pair of the integro-differential Equations (69) if we extract $W(x)$ from the second Equation (69), put it into the first one, and fulfill the appropriate differentiation. The key equation has the form of the Euler equation

$$x^3 E''' + x^2 E''(6+\nu+2\omega_0) + xE'(3K_1+4+5\omega_0+4\nu+2\nu\omega_0) + 3E[K_1(3+\omega_0)+\nu\omega_0] = 0. \tag{70}$$

The corresponding characteristic equation

$$\sigma^3 + \sigma^2(3+\nu+2\omega_0) + \sigma(3K_1+3\omega_0+3\nu+2\nu\omega_0) + 3[K_1(3+\omega_0)+\nu\omega_0] = 0 \tag{71}$$

has three coinciding roots $\sigma_1 = \sigma_2 = \sigma_3 = 0$, when

$$\nu = -(3+2\omega_0), \quad K_1 = \frac{\omega_0(3+2\omega_0)}{(3+\omega_0)}, \tag{72}$$

and $\omega_0$ satisfies the following cubic equation:

$$4\omega_0^3 + 15\omega_0^2 + 27\omega_0 + 27 = 0. \tag{73}$$

This equation has only one real root $\omega_0 \approx -2.06$. This unique set of parameters gives us the submodel with the following functions $E(x)$ and $W(x)$:

$$E(x) = E(x_*) + x_* E'(x_*) \log\left(\frac{x}{x_*}\right) + \tilde{C}_3 \log^2\left(\frac{x}{x_*}\right), \tag{74}$$

$$W(x) = W(x_*) + x_* W'(x_*) \log\left(\frac{x}{x_*}\right) + \tilde{C}_3 \frac{(3+\omega_0)}{\omega_0} \log^2\left(\frac{x}{x_*}\right). \tag{75}$$

The modified initial data at $t = t_*$ ($x = x_*$) can be obtained from (69); they now have the form:

$$x_* W'(x_*) = \omega_0[E(x_*) - W(x_*)], \quad x_* E'(x_*) = \omega_0 W(x_*) - (3+\omega_0)E(x_*), \tag{76}$$

so that the constant $\tilde{C}_3$ can be presented in two equivalent versions:

$$2\tilde{C}_3 = \omega_0 x_* W'(x_*) - (3+\omega_0)x_* E'(x_*) = E(x_*)\left[\omega_0^2 + (3+\omega_0)^2\right] - \omega_0 W(x_*)(3+2\omega_0). \tag{77}$$

The square of the Hubble function can be presented as follows:

$$\frac{3}{\kappa_*}\left[H^2(x) - H^2(x_*)\right] =$$

$$-3E(x_*)\log\left(\frac{x}{x_*}\right) + \left(\frac{3+2\omega_0}{2\omega_0}\right)\left[E(x_*)\left(2\omega_0^2+6\omega_0+9\right) - W(x_*)\omega_0(3+2\omega_0)\right]\log^2\left(\frac{x}{x_*}\right), \tag{78}$$

where we used the following definition:

$$H^2(x_*) = \frac{\Lambda_*}{3} + \frac{\kappa_*}{3}[W(x_*) + E(x_*)]. \tag{79}$$

The principal detail of our model is that the aether fixes the moment $t_*$ (and thus the value $x_*$) so that the coefficient in front of $\log^2\left(\frac{x}{x_*}\right)$ in Formula (78) is negative. Keeping in mind that $E(x_*)$ and $W(x_*)$ are fixed by the Formulas (61) and (60) with $t = t_*$, respectively, we see that this is possible when

$$0.41 < \frac{\sqrt{3}H(t_0)}{\sqrt{\kappa_* W(t_0)}} \sinh\left[\gamma\sqrt{3\kappa_* W(t_0)}(t_*-t_0)\right] < 4.91. \tag{80}$$

(Here, we used $\omega_0 \approx -2.06$). Since the hyperbolic sinus is the monotonic function, we can find the appropriate value $t_* > t_0$ for every set of parameters $H(t_0) > 0$, $\kappa_* > 0$, $W(t_0) > 0$, and $\gamma > 0$.

### 3.3.2. Exact Solutions in the Periodic Model

In order to find the scale factor and the Hubble function for the mentioned case, we can use the simplified method. We introduce the auxiliary definitions

$$\alpha^2 = -\kappa_* \left(\frac{3+2\omega_0}{6\omega_0}\right) \left[E(x_*)\left(2\omega_0^2+6\omega_0+9\right)-W(x_*)\omega_0(3+2\omega_0)\right], \quad 2\beta = \kappa_* E(x_*), \quad (81)$$

and search for the $a(t)$ and $H(t)$ in the form

$$a(t) = a_X \exp\left\{h_X \sin\left[\alpha(t-t_*)+\varphi\right]\right\}, \quad a_X = a(x_*) \exp\left(-\frac{\beta}{\alpha^2}\right), \quad (82)$$

$$H(t) = \frac{\dot{a}}{a} = \alpha h_X \cos\left[\alpha(t-t_*)+\varphi\right]. \quad (83)$$

Then, we put these quantities into (78) and obtain the relationship between the introduced parameters

$$\alpha^2 h_X^2 = H^2(t_*) + \frac{\beta^2}{\alpha^2}. \quad (84)$$

The last point is to find the auxiliary parameter $\varphi$ which plays the role of a phase of the metric oscillations. We require that at $t = t_*$, the following initial condition takes place

$$a(t_*) = a_X \exp\left[h_X \sin\varphi\right], \quad H(t_*) = \alpha h_X \cos\varphi, \quad (85)$$

and thus

$$\sin\varphi = \frac{\beta}{h_X\alpha^2} \leq 1, \quad \cos\varphi = \frac{H(t_*)}{\alpha h_X} \leq 1 \ \Rightarrow \tan\varphi = \frac{\beta}{H(t_*)\alpha}. \quad (86)$$

Illustration of the behavior of the scale factor is presented on Figure 3.

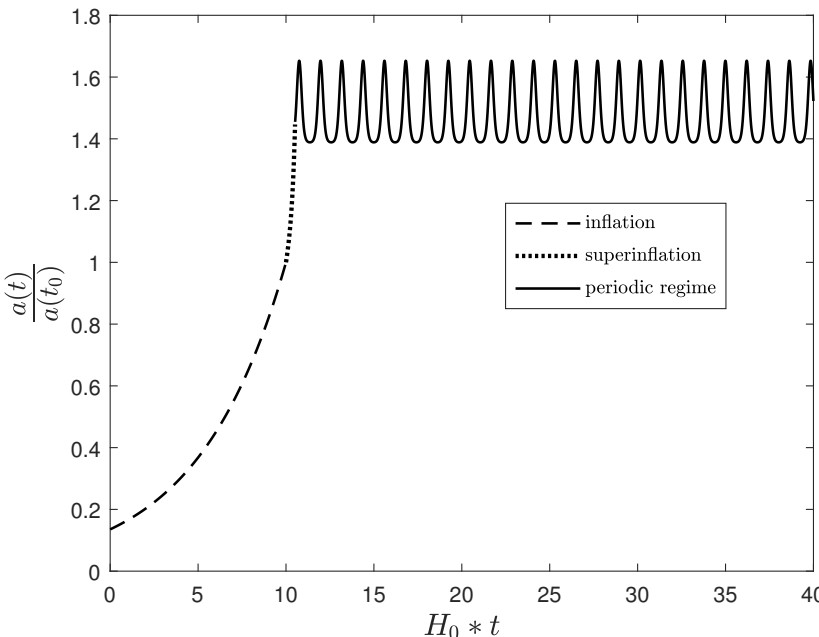

**Figure 3.** Illustration of the behavior of the scale factor during the periodic episode of the Universe evolution. The interval of time $t < t_0$ relates to the epoch of the quantum regime of the expansion; near the moment $t = t_0$, the ratio $\frac{a(t_0)}{a(0)}$ reached the value of the order $10^{26}$. During the interval $t_0 < t < t_*$, the fine-tuning provided by the dynamic aether took place, and at $t > t_*$, we deal with the oscillatory regime.

This exact solution demonstrates that the Hubble function is periodic with a period $T = \frac{2\pi}{\alpha}$; the parameter $\alpha$ plays the role of the frequency of the oscillations of the spacetime. The energy densities of the non-axionic dark matter and dark energy are also periodic:

$$E(t) = E_X + \frac{6\omega_0\alpha^2 h_X}{\kappa_*(3+2\omega_0)^2} \sin\left[\alpha(t-t_*)+\varphi\right] - \frac{3\omega_0\alpha^2 h_X^2}{\kappa_*(3+2\omega_0)} \sin^2\left[\alpha(t-t_*)+\varphi\right], \quad (87)$$

$$W(t) = W_X - \frac{6\omega_0\alpha^2 h_X}{\kappa_*(3+2\omega_0)^2} \sin\left[\alpha(t-t_*)+\varphi\right] - \frac{3\alpha^2 h_X^2(3+\omega_0)}{\kappa_*(3+2\omega_0)} \sin^2\left[\alpha(t-t_*)+\varphi\right]. \quad (88)$$

Here, for the sake of convenience, we marked the values of the energies at the moment when the argument of sinus takes a zero value by $E_X$ and $W_X$. The energy densities of the dark energy (88) and of the non-axionic dark matter (87) describe oscillations not only with the frequency $\alpha$, but also with the double frequency $2\alpha$.

Finally, one can find the guiding function $\Phi_*(t)$ in the integral form

$$\Phi_*(t) = \Phi_*(t_*) + \dot{\Phi}_*(t_*)\frac{1}{\alpha} e^{\frac{3\beta}{2\alpha^2}} \int_{\tau_1}^{\tau_2} d\tau e^{-\frac{3}{2}h_X \cos\tau}, \quad (89)$$

where

$$\tau_2 = \tau_1 + \alpha(t-t_*), \quad \tau_1 = \arctan\left[\frac{\beta}{\alpha H(t_*)}\right] - \frac{\pi}{2}. \quad (90)$$

Keeping in mind the representation of the complete Bessel function of the imaginary argument (see Section 6.22 (4) of the book [33]).

$$I_0(z) = \frac{1}{\pi} \int_0^\pi d\xi e^{z\cos\xi}, \quad (91)$$

we could rewrite the integral in (89) as

$$\int_{\tau_1}^{\tau_2} d\tau e^{-\frac{3}{2}h_X \cos\tau} = \tilde{I}_0\left(-\frac{3}{2}h_X, \tau_2\right) - \tilde{I}_0\left(-\frac{3}{2}h_X, \tau_1\right), \quad \tilde{I}_0(z,\tau) = \frac{1}{\pi}\int_0^\tau d\xi e^{z\cos\xi}, \quad (92)$$

using the incomplete Bessel function $\tilde{I}_0(z,\tau)$. However, these mathematical details are out from the frame of this article.

### 3.3.3. Parametric Generation of the Axion Field

Let us return to the perturbed equations for the axion field evolution (see (63) and (64)), but now, we use the periodic Hubble function (83) and the transformation $\psi \to \left[\frac{a(t)}{a(t_*)}\right]^{-\frac{3}{2}}\Psi$ based on the scale factor (82). Now, we deal with the standard Hill equation

$$\ddot{\Psi} = \mathcal{J}(t)\Psi, \quad (93)$$

with the periodic Hill potential

$$\mathcal{J}(t) = \left[\frac{9}{8}H^2(t_X) - m_A^2\right] - \frac{3}{2}\alpha H(t_X)\sin\left[\alpha(t-t_*)+\varphi\right] + \frac{9}{8}H^2(t_X)\cos\left[2\alpha(t-t_*)+2\varphi\right]. \quad (94)$$

The Hill potential is characterized by the period $T = \frac{2\pi}{\alpha}$. The Wronsky determinant for the Equation (93) is equal to a constant. As usual, we work with the reduced dimensionless time variable $\tau = \alpha(t-t_*)+\varphi$ and consider the starting point to be $\tau = 0$. In these terms, the Hill potential possesses the period $2\pi$. It is convenient to consider the fundamental solutions $\Psi_1(\tau)$ and $\Psi_2(\tau)$ which satisfy the initial data

$$\Psi_1(0) = 1, \quad \Psi_1'(0) = 0, \quad \Psi_2(0) = 0, \quad \Psi_2'(0) = 1, \quad (95)$$

where the prime denotes the derivative with respect to $\tau$. For such fundamental solutions, the Wronsky determinant is equal to 1. If $\Psi_1(\tau)$ and $\Psi_2(\tau)$ satisfy the Hill equations, the functions $\Psi_1(\tau + 2\pi)$ and $\Psi_2(\tau + 2\pi)$ are also the solutions to the Hill equation with the Hill potential possessing the property $\mathcal{J}(\tau + 2\pi) = \mathcal{J}(\tau)$. According to the Floquet theorem [34], we know that for the periodic Hill potential, there exists the so-called normal solution $\Psi_N$, which satisfies the rule $\Psi_N(\tau + 2\pi) = \sigma \Psi_N(\tau)$. The parameter $\sigma$ is the solution to the characteristic equation

$$\sigma^2 - 2\mathcal{A}\sigma + 1 = 0 \;\Rightarrow\; \frac{1}{2}\left(\sigma + \frac{1}{\sigma}\right) = \mathcal{A}, \tag{96}$$

where $2\mathcal{A}$ denotes the trace of the Wronsky matrix calculated at the end of the first period

$$\mathcal{A} = \frac{1}{2}\left[\Psi_1(2\pi) + \Psi_2'(2\pi)\right]. \tag{97}$$

Then, we represent the characteristic number $\sigma$ as an exponent $\sigma = e^{2\pi\mu}$ providing the characteristic equation to have the form $\cosh 2\pi\mu = \mathcal{A}$. Clearly, when $\mathcal{A} > 1$, there are two real solutions to the characteristic equation

$$\sigma_1 = \mathcal{A} + \sqrt{\mathcal{A}^2 - 1} = e^{2\pi\mu}, \quad \sigma_2 = \mathcal{A} - \sqrt{\mathcal{A}^2 - 1} = e^{-2\pi\mu}, \quad \sigma_1\sigma_2 = 1, \tag{98}$$

$$2\pi\mu = \log\left(\mathcal{A} + \sqrt{\mathcal{A}^2 - 1}\right). \tag{99}$$

Finally, the equation $\Psi_N(\tau + 2\pi) = e^{2\pi\mu}\Psi_N(\tau)$ can be rewritten as

$$e^{-\mu(\tau + 2\pi)}\Psi_N(\tau + 2\pi) = e^{-\mu\tau}\Psi_N(\tau), \tag{100}$$

and we see that $e^{-\mu\tau}\Psi_N(\tau)$ is a periodic function, and thus $\Psi_N(\tau) = e^{\mu\tau}Y(\tau)$, where $Y(\tau)$ denotes a periodic function. Thus, following the idea of the Floquet theorem, we obtain that one of the solutions to the Hill equation grows exponentially, if the characteristic Equation (96) has two different real roots. The periodic function $Y(\tau)$ can be obtained as follows. We put the function $\Psi_N(\tau) = e^{\mu\tau}Y(\tau)$ into the Equation (93) and obtain the equation for $Y(\tau)$ in the form

$$Y'' + 2\mu Y' = \left(\frac{\mathcal{J}}{\alpha^2} - \mu^2\right)Y. \tag{101}$$

Then, we search for periodic function $Y(\tau)$ in the form of trigonometric series

$$Y(\tau) = f_0 + \sum_{k=1}^{\infty}\left[f_k \sin k\tau + g_k \cos k\tau\right]. \tag{102}$$

It is convenient to use now the auxiliary terms

$$F_0 = \frac{9H^2(t_X)}{8\alpha^2} - \frac{m_A^2}{\alpha^2} - \mu^2, \quad F_1 = -\frac{3H(t_X)}{2\alpha}, \quad F_2 = \frac{9H^2(t_X)}{8\alpha^2}, \tag{103}$$

in order to formulate the recurrent relations between the coefficients $f_0$, $f_k$, and $g_k$. These recurrent relations contain four subgroups: for $k = 0$, $k = 1$, $k = 2$, and $k \geq 3$:

$$k = 0 \;\Rightarrow\; F_0 f_0 + \frac{1}{2}F_1 f_1 + \frac{1}{2}F_2 g_2 = 0, \tag{104}$$

$$k = 1 \;\Rightarrow\; f_0 F_1 + f_1\left(1 + F_0 - \frac{1}{2}F_2\right) - 2\mu g_1 - \frac{1}{2}F_1 g_2 + \frac{1}{2}F_2 f_3 = 0, \tag{105}$$

$$k = 1 \;\Rightarrow\; g_1\left(1 + F_0 + \frac{1}{2}F_2\right) + 2\mu f_1 + \frac{1}{2}F_1 f_2 + \frac{1}{2}F_2 g_3 = 0, \tag{106}$$

$$k = 2 \;\Rightarrow\; f_2(4 + F_0) - 4\mu g_2 + \frac{1}{2}F_1(g_1 - g_3) + \frac{1}{2}F_2 f_4 = 0\,, \tag{107}$$

$$k = 2 \;\Rightarrow\; g_2(4 + F_0) + 4\mu f_2 + F_2 f_0 + \frac{1}{2}F_1(f_3 - f_1) + \frac{1}{2}F_2 g_4 = 0\,, \tag{108}$$

$$k \geq 3 \;\Rightarrow\; (k^2 + F_0)f_k + 2\mu k g_k + \frac{1}{2}F_1(g_{k-1} - g_{k+1}) + \frac{1}{2}F_2(f_{k+2} + f_{k-2}) = 0\,, \tag{109}$$

$$k \geq 3 \;\Rightarrow\; (k^2 + F_0)g_k - 2\mu k f_k + \frac{1}{2}F_1(f_{k+1} - f_{k-1}) + \frac{1}{2}F_2(g_{k+2} + g_{k-2}) = 0\,. \tag{110}$$

Clearly, there is the consistent sequential procedure: we extract $g_{k+2}$ from (110), $f_{k+2}$ from (109), $g_4$ from (108), $f_4$ from (107), $g_3$ from (106), $f_3$ from (105), and $g_2$ from (104), thus allowing the coefficients $f_2$, $g_1$, $f_1$, and $f_0$ to remain arbitrary. As an illustration of the instable solution, we can put $g_1 = 0$, $f_2 = 0$ and express all the coefficients via the parameters $f_0$ and $f_1$. The corresponding decomposition has the form

$$\Psi_N(\tau) = e^{\mu\tau}\left\{ f_0 + f_1\sin\tau - \frac{1}{F_2}(2F_0 f_0 + F_1 f_1)\cos 2\tau - \right.$$

$$\left. -\frac{2}{F_2}\left[F_1 f_0\left(1 + \frac{F_0}{F_2}\right) + f_1\left(2 + F_0 - \frac{1}{2}F_2\right)\right]\sin 3\tau - \frac{4\mu}{F_2}f_1\cos 3\tau + \ldots\right\}. \tag{111}$$

The coefficients $f_0$ and $f_1$ can be formally found from the initial data

$$\Psi_N(0) = f_0 + \sum_{k=1}^{\infty} g_k\,, \quad \Psi_N'(0) = \mu\Psi_N(0) + \sum_{k=1}^{\infty} k f_k\,. \tag{112}$$

To conclude, during the episode of the Universe evolution, when the scale factor is the periodic function (82), one of the solutions to the perturbed equation for the axion field behaves as

$$\psi(\tau) = Y(\tau)\exp\left\{\mu\tau - \frac{3}{2}h_X\sin\tau\right\}. \tag{113}$$

The envelope function $\exp\left\{\mu\tau - \frac{3}{2}h_X\sin\tau\right\}$ is the sinusoidally deformed exponent which grows during the periodic episode of Universe evolution (see Figure 4 for illustration). We deal with the instability of the axion field, which allows us to speak about an "axionization" of the Universe in analogy with the process indicated as scalarization in the work [35].

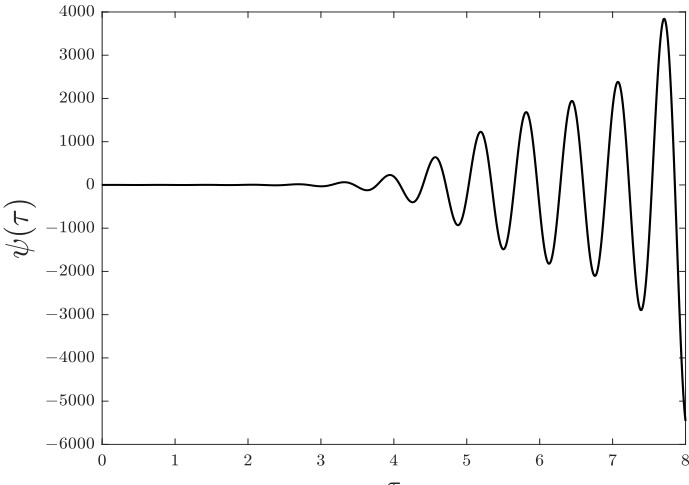

**Figure 4.** Illustration of the behavior of the axion perturbation function $\psi(\tau)$ in the regime of parametric excitation.

### 3.4. Remark about the Post-Inflationary Stage of the Universe Evolution

When we consider the model of aetheric control over the dark fluid evolution, we assume that there exists a critical value of the expansion scalar, say, $\Theta_{**} = 3H(t_{**})$, so that for $t > t_{**}$, the rheological interactions in the dark fluid happen to be switched off, and we deal with the post-inflationary epoch. For this epoch, the dark energy and non-axionic dark matter are assumed to be characterized by the standard equations of state $P = -W$ and $\Pi = 0$, respectively. Also, $\omega_0 = 0$, and the dark matter interacts with dark energy via the gravity field only. The conservation laws for the dark energy and non-axionic dark matter become decoupled, and we now have

$$\dot{W} = 0, \quad \dot{E} + 3HE = 0 \;\Rightarrow W = const = W(t_{**}), \quad E(t) = E(t_{**})\left[\frac{a(t_{**})}{a(t)}\right]^3. \tag{114}$$

As for the axion field, we assume that during the period of instability, this field reached the maximal value at $t = t_{(\max)}$, and at the moment $t = t_{**} > t_{(\max)}$, two parameters, $\phi(t_{**})$ and $\dot{\phi}(t_{**})$, start to play the roles of the initial data for the master equations in the post-inflationary period. Clearly, we can recover all the specific details of the standard post-inflationary epochs, but here, we mention two interesting cases of the axion field behavior at $t > t_{**}$.

### 3.4.1. Frozen Axion Field with Non-Vanishing Effective Cosmological Constant

If the value $\phi(t_{**})$ happens to be close to one of the minima of the axion field potential, i.e., $\phi(t_{**}) \to n_{**}\Phi_*$ with some integer $n_{**}$ and thus $V \to 0$, we obtain the situation analogous to the one described in [36], when the large kinetic energy of axions dominates its potential energy, so the axion field evolves as a stiff matter. Now, one can see that the contribution of the derivative of the potential also is small, and the equation of axion dynamics (33) admits the integral

$$\dot{\phi} = \dot{\phi}(t_{**})\left[\frac{a(t_{**})}{a(t)}\right]^3. \tag{115}$$

Thus, the key equation of the gravity field takes the form

$$3H^2\left(1 + \frac{3}{2}C_2\right) = \Lambda + \kappa W(t_{**}) + \frac{1}{2}\kappa\Psi_0^2(\dot{\phi}(t_{**}))^2\left[\frac{a(t_{**})}{a(t)}\right]^6 + \kappa E(t_{**})\left[\frac{a(t_{**})}{a(t)}\right]^3. \tag{116}$$

If the effective cosmological constant $\Lambda + \kappa W(t_{**})$ is non-vanishing, we can rewrite this equation in terms of auxiliary variable $x = \frac{a(t)}{a(t_{**})}$ as

$$H^2 = H_\infty^2\left[1 + 2\mathcal{M}_1 x^{-3} + \mathcal{M}_2 x^{-6}\right], \tag{117}$$

where the following auxiliary parameters are introduced:

$$H_\infty^2 = \frac{\Lambda + \kappa W(t_{**})}{3(1 + \frac{3}{2}C_2)}, \quad 2\mathcal{M}_1 H_\infty^2 = \frac{\kappa E(t_{**})}{3(1 + \frac{3}{2}C_2)}, \quad \mathcal{M}_2 H_\infty^2 = \frac{\kappa\Psi_0^2(\dot{\phi}(t_{**}))^2}{6(1 + \frac{3}{2}C_2)}. \tag{118}$$

The solution to the equation

$$H_\infty(t - t_{**}) = \int_1^{\frac{a(t)}{a(t_{**})}} \frac{dx}{x\sqrt{1 + 2\mathcal{M}_1 x^{-3} + \mathcal{M}_2 x^{-6}}} \tag{119}$$

can be presented in the form

$$\frac{a(t)}{a(t_{**})} = \left\{(1 + \mathcal{M}_1)\cosh\left[3H_\infty(t - t_{**})\right] - \mathcal{M}_1 + \sqrt{1 + 2\mathcal{M}_1 + \mathcal{M}_2}\,\sinh\left[3H_\infty(t - t_{**})\right]\right\}^{\frac{1}{3}}. \tag{120}$$

In the asymptotic regime $t \to \infty$, we obtain that $H \to H_\infty$ and $a(t) \propto e^{H_\infty t}$, i.e., we deal with the de Sitter type asymptote.

### 3.4.2. Frozen Axion Field with Vanishing Effective Cosmological Constant

When $\Lambda + \kappa W(t_{**}) = 0$, we use another re-parametrization of the key equation for the gravity field:

$$H^2 = \rho_1 x^{-3} + \rho_2 x^{-6}, \quad \rho_1 = \frac{\kappa E(t_{**})}{3(1 + \frac{3}{2}C_2)}, \quad \rho_2 = \frac{\kappa \Psi_0^2 (\dot\phi(t_{**}))^2}{6(1 + \frac{3}{2}C_2)}. \tag{121}$$

The solution for the scale factor is now of the form

$$\frac{a(t)}{a(t_{**})} = \left[1 + 3\sqrt{\rho_1 + \rho_2}(t - t_{**}) + \frac{9}{4}\rho_1 (t - t_{**})^2\right]^{\frac{1}{3}}. \tag{122}$$

In the asymptotic regime, we obtain the power-law behavior of the scale factor, $a(t) \propto t^{\frac{2}{3}}$.

## 4. Discussion and Conclusions

Starting from 1996, cosmologists use specific terminology which is associated with the spontaneous growth of fields and particle numbers in the early Universe. These terms are spontaneous scalarization (see, e.g., [35,37,38]), vectorization [39,40], tensorization [41], and spinorization [42]. Thinking along this line, we pose the question: when and why did the axionization of the Universe occur? In other words, we are interested to elaborate upon the models of spontaneous growth of axions which form the main part of the cosmic dark matter. There are two mechanisms of such spontaneous growth: the first one is associated with axion production due to the decay of gauge fields (see, e.g., [43]); the second (the most known) mechanism is connected with the instability of the axion system. In this work, we consider models of the second type and assume that the origin of the axion instability is connected with the rate of the Universe evolution. We can explain the idea using a few examples. The first example relates to standard inflation, when the scale factor is described by the de Sitter law $a(t) \propto e^{H_0(t - t_0)}$ with $H(t) = H_0 = const$. Equation (63) is now a differential equation of second order with constant coefficients, and we conclude that both fundamental solutions to this equation do not increase; the model is stable with respect to perturbations. The second example is connected with the power-law behavior of the scale factor $a(t) \propto t^\xi$ (it can be obtained, e.g., for the Universe filled with matter with the equation of state $P = (\Gamma - 1)W$). Equation (63) can be now reduced to the Bessel equation, and if $\Gamma \leq 2$, the fundamental solutions to this equation also do not increase, so the model is stable. The third example is the super-inflationary solution obtained in this work. As we see from Formula (65) and from Figure 2, there exists the interval of the cosmological time when the axion field increases, such that the axion system should be considered instable. The fourth example is the model with periodic behavior of the scale factor (see (82)). Now, we see from (113) and from Figure 4 that the perturbations of the axion field grow during the episode of periodic evolution of the Universe. Again, we deal with axionic instability, and thus, we can speak about Universe axionization.

To conclude, we have to say that at present, we do not know certainly that the super-inflationary and/or oscillatory periods took place in the history of the early Universe. Systematic analysis of observational data is necessary to clarify this question, and we hope to take part in the detailed discussions. As a first step, one could try to think about imprints that were left by the oscillations of the scale factor in the early-time and late-time stages of Universe evolution. For sure, the main features relate to the axion field growth in the early Universe and formation of the cold dark matter from these relic axions in our epoch. Of course, it would be important to consider a model which unifies these features along the lines elaborated, e.g., in [44–47]; we hope to present this work in our next papers.

The mechanism of the axion field growth described by the periodic model can be indicated as the parametric mechanism of the Floquet type; this mechanism is well known in nonlinear mechanics. In the work [48], the parametric generation of the scalar field was studied in the case when the oscillations of the scalar field are intrinsic, i.e., they are connected with the properties of the scalar field potential. Our approach is based on the idea that just the oscillations of the gravity field are the cause of parametric instability of the pseudoscalar axion field. As for the model of super-inflation studied in this work, the mechanism of axion instability can be indicated by a generalized parametric model, since instead of the standard Hill equation, we use the generalized Hill equation for the analysis, replacing the trigonometric functions with hyperbolic ones.

We have to emphasize that in all the stages of the Universe evolution, Peccei–Quinn symmetry is assumed to be non-violated; the discrete symmetry $\phi \to \phi + m\Phi_*$ with integer $m$ is supported in our model due to the specific periodic structure of the axion field potential (3).

The last point of our discussion is the status of two exact solutions to the master equations of the presented model: the super-inflationary and the periodic solutions. These exact solutions appear in the model with rheologically active dark fluid consisting of coupled dark energy and non-axionic dark matter. These solutions are marked in the catalogs presented in [9,10] as the ones corresponding to the triple degenerate roots of the characteristic equation. In this paper, we considered both solutions in detail and studied a new composite model in which the super-inflationary behavior of the Universe performs a fine-tuning step, allowing the start of the periodic regime. In this context, we have to emphasize that such a behavior of the expanding Universe is possible due to the aetheric guidance, which is described in our model by inclusion of the expansion scalar $\Theta$ into the axion field potential and into the Lagrangian of the dark fluid. Also, we believe that namely the dynamic aether predetermines the milestone values of the cosmological time $t_0$, $t_*$, as well as the time moment $t_{**}$, when the periodic episode of the Universe evolution finishes and the standard cosmological expansion starts.

**Author Contributions:** Conceptualization, A.B.; methodology, A.I.; software, A.S.; validation, A.S. and A.I.; formal analysis, A.S.; investigation, A.B., A.S. and A.I.; resources, A.I.; data curation, A.S.; writing—original draft preparation, A.B.; writing—review and editing, A.B., A.S. and A.I.; visualization, A.S.; supervision, A.B.; All authors have read and agreed to the published version of the manuscript.

**Funding:** Russian Foundation for Basic Research (Grant No. 20-52-05009).

**Data Availability Statement:** Not applicable.

**Acknowledgments:** The work was supported by the Russian Foundation for Basic Research (Grant N 20-52-05009)

**Conflicts of Interest:** The authors declare no conflict of interest.

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
