# Peer review of "Interaction of the Cosmic Dark Fluid with Dynamic Aether: Parametric Mechanism of Axion Generation in the Early Universe"

_symmetry, doi:10.3390/sym15101824_

Round 1

Reviewer 1 Report

Below are my comments to the paper:

  1. The paper motivates the super inflation as a longer inflationary period due to the fact that it allows longer macroscopic causal interactions. However, the causality issue can already be explained in the standard inflation. Are there other reasons to introduce the super inflation?
  1. Does the dynamic aether have non-vanishing vacuum expectation values? if so, do they violate isotropy? 
  1. In line 278, the authors mention that the guiding functions can be replaced by axions, are there further assumptions that one can do this replacement?
  1. Given the oscillatory feature in the scale factor during the super-inflationary stage, what imprints do they leave on the early-time and late-time observables?
  1. The conclusion is somewhat misleading in stating that the axions are fraction for dark matter.

the paper is well written.

Author Response

Dear Collegue, thank you very much for valuable remarks and for your high assessment of our work.

  1. The paper motivates the super inflation as a longer inflationary period due to the fact that it allows longer macroscopic causal interactions. However, the causality issue can already be explained in the standard inflation. Are there other reasons to introduce the super inflation?”

Answering this comment we added the following paragraph (lines 34-53)

{\bf Such a possibility appears, for instance, if we consider the super-inflation. The term super-inflation has been already used, e.g., in the model of Loop Quantum Cosmology \cite{LQC} in order to mark a specific episode of the Universe inflationary evolution, when the kinetic energy of the scalar field is much more that the potential energy. Our goal is to consider not approximate but exact solutions describing the super-inflation as an alternative to the standard inflation. Of course, the inflation scenario has already explained many details of the early Universe evolution, and when we pose a question about a super-inflation, we have to motivate this step. Keeping in mind this simple argument, we would like to attract attention to one new fact only. The LIGO - VIRGO Collaboration has proved that black holes with intermediate masses (from 20 to 90 solar masses) do exist. Astrophysicists are ready to explain theoretically the presence of black holes with masses of several solar masses obtained in the scenario of a star collapse; also, one can explain the existence of super-massive black holes. However, now there is no adequate theory for the formation of the medium-sized black holes and super-massive stars. But if the causal period of the early Universe evolution lasted longer than the inflation theory predicts, we have a natural opportunity to explain the observed set of the black hole masses. We hope that a corresponding model for the formation of the medium-sized black holes will be formulated in the near future, for example, similar to how the problem of the causal limit of the neutron star maximum mass was solved in \cite{CausalLimit}.}

  1. “Does the dynamic aether have non-vanishing vacuum expectation values? if so, do they violate isotropy?

Answering this comment we added the following paragraph (lines 194-200)

{\bf If the guiding function $\Phi_*$ is constant, the more convenient term can be used for this quantity, namely, the vacuum expectation value. In this regard, it is important to mention that, unlike the axion theory, the standard theory of the dynamic aether does not contain an appropriate vector field potential. In this sense the vacuum expectation value of the vector field does not appear in the standard version of this theory, and thus, there is no fixed direction in the space, and the spatial isotropy violation can not take place.}

  1. “In line 278, the authors mention that the guiding functions can be replaced by axions, are there further assumptions that one can do this replacement?”

In order to clarify this detail we added the following paragraph (lines 294-299)

{\bf Let us imagine that the axion field $\phi$ is frozen in the first minimum of the axion potential; it coincides with the guiding function and thus, it follows the variations of $\Phi_*(t)$. Then we put $\phi {=} \Phi_*$ into (\ref{N24}) and we have to admit that the axion field remains in the first minimum, if the guiding function $\Phi_*$ satisfies the equation

\begin{equation}

\ddot{\Phi}_* + 3H \dot{\Phi}_* = 0 \,.

\label{N234}

\end{equation}

Our ansatz is that (\ref{N234}) defines the missing master equation for the guiding function $\Phi_*$ in the framework of the established model. }

  1. “Given the oscillatory feature in the scale factor during the super-inflationary stage, what imprints do they leave on the early-time and late-time observables?”

We have written the following paragraph trying to comment this remark (lines 572-581):

{\bf To conclude, we have to say, that at present we do not know certainly that the super-inflationary and/or oscillatory periods took place in the history of the early Universe. Systematic analysis of observational data is necessary to clarify this question, and we hope to take part in the detailed discussions. As a first step, one could try to think about imprints that were left by the oscillations of the scale factor in the early-time and late-time stages of the Universe evolution. For sure, the main features relate to the axion field growth in the early Universe, and formation of the cold dark matter from these relic axions in our epoch. Of course, it would be important to consider the model, which unifies these features along the lines elaborated, e.g., in \cite{U1,U2,U3,U4}; we hope to do this work in the next papers.}

  1. “The conclusion is somewhat misleading in stating that the axions are fraction for dark matter”

 In order to eliminate some misleading in this point, we have replaced the term “fraction” by “axion component” or “axion part of dark matter”. Also, we add the phrase (see lines 82-86):

{\bf We separate the axionic component of the dark matter \cite{AX}, and consider axions on the language of field theory, as a pseudoscalar field with modified periodic potential. Other components of the dark matter (WIMPs, ALPs, warm and hot dark matter parts, etc.) are unified and described as a dark medium with rheological properties.}

Reviewer 2 Report

This work is devoted to an unconventional mixture towards explaining the early Universe cosmology. The components that come to play are the aether fluid, the axionic component of dark matter and other dark matter components in terms of fluids which interact. I have several comments. Why do the authors chose this specific 5 component model to describe the early Universe and why did they chose dark matter to composed partially from axions and not entirelly? The result is a questionable super-inflation evolution, known also to occur in Loop Quanum Cosmology frameworks, and an oscillating period of the Universe post-inflationary, which is also not certain that it occured. Also in standard axion frameworks, in the context of modified gravity, the misalignment axion (cite 1510.07633) is used in which the primordial U(1) Peccei-Quinn symmetry is broken, see for example 2012.00586,2208.05544, for some recent relevant works. In this framework the axion is basically frozen during inflation working its way to the bottom of the potential, in which case modified gravity controls inflation and then the axion once it is in the bottom, it starts its oscillations and it redshifts as dark matter. Solely the axion oscillates though, not the Universe. So my question is, is the Peccei Quinn symmetry broken in the case of the model that the authors use? It probably is due to the residual shift symmetry preserving cosine potential (natural inflation). Thus instead of a natural inflation-like inflationary era, a super-inflation era commences followed by oscillations. It seems that the synergy of axion and aether causes this, but the authors need to clarify this issue, comparing their results with the aforementioned scenarios. Also, does the axion redshift as standard dark matter post-inflationary in the context of the model that the authors develop? Some comments and discussions on this could be useful. After these amendments the article can be accepted for publication.

Author Response

Dear Collegue, thank you very much for valuable remarks and for your high assessment of our work.

  1. “Why do the authors chose this specific 5 component model to describe the early Universe and why did they chose dark matter to compose partially from axions and not entirelly?”

Answering this comment, we kept in mind that till now the light axions are considered to be “hypothetical”, and there is a number of experimantal groups, which try to detect more massive particles (WIMPs and ALPs). That is why we have restricted ourselves by the phrase (see lines 82-86):

{\bf We separate the axionic component of the dark matter \cite{AX}, and consider axions on the language of field theory, as a pseudoscalar field with modified periodic potential. Other components of the dark matter (WIMPs, ALPs, warm and hot dark matter parts, etc.) are unified and described as a dark medium with rheological properties.}

 “The result is a questionable super-inflation evolution, known also to occur in Loop Quanum Cosmology frameworks, and an oscillating period of the Universe post-inflationary, which is also not certain that it occured.”

In order to comment this remark we, first, added the phrase (see lines 34-39):

The term super-inflation has been already used, e.g., in the model of Loop Quantum Cosmology \cite{LQC} in order to mark a specific episode of the Universe inflationary evolution, when the kinetic energy of the scalar field is much more that the potential energy. Our goal is to consider not approximate but exact solutions describing the super-inflation as an alternative to the standard inflation.

Also, in the lines 572-581 we added the following paragraph:

{\bf To conclude, we have to say, that at present we do not know certainly that the super-inflationary and/or oscillatory periods took place in the history of the early Universe. Systematic analysis of observational data is necessary to clarify this question, and we hope to take part in the detailed discussions. As a first step, one could try to think about imprints that were left by the oscillations of the scale factor in the early-time and late-time stages of the Universe evolution. For sure, the main features relate to the axion field growth in the early Universe, and formation of the cold dark matter from these relic axions in our epoch. Of course, it would be important to consider the model, which unifies these features along the lines elaborated, e.g., in \cite{U1,U2,U3,U4}; we hope to do this work in the next papers.}

  1. Also in standard axion frameworks, in the context of modified gravity, the misalignment axion (cite 1510.07633) is used in which the primordial U(1) Peccei-Quinn symmetry is broken, see for example 2012.00586,2208.05544, for some recent relevant works. In this framework the axion is basically frozen during inflation working its way to the bottom of the potential, in which case modified gravity controls inflation and then the axion once it is in the bottom, it starts its oscillations and it redshifts as dark matter. Solely the axion oscillates though, not the Universe. So my question is, is the Peccei Quinn symmetry broken in the case of the model that the authors use? It probably is due to the residual shift symmetry preserving cosine potential (natural inflation). Thus instead of a natural inflation-like inflationary era, a super-inflation era commences followed by oscillations. It seems that the synergy of axion and aether causes this, but the authors need to clarify this issue, comparing their results with the aforementioned scenarios.

Also, does the axion redshift as standard dark matter post-inflationary in the context of the model that the authors develop? Some comments and discussions on this could be useful. “

We are grateful to the Referee for this valuable remark.

In order to continue this part of discussion we have written a new Subsection 3.4. (see lines 509-543, the formulas (114)-(122) and new references [35],[44-47]).

Concerning the point about the Peccei-Quinn symmetry violation, we added the phrase (lines 592-595):

{\bf We have to emphasize that in all the stages of the Universe evolution the Peccei-Quinn symmetry is assumed to be non-violated; the discrete symmetry $\phi \to \phi + m \Phi_*$ with integer $m$ is supported in our model due to the specific periodic structure of the axion field potential (\ref{13})}.

Reviewer 3 Report

I have reproduced the results in the paper, which seem correct from my perspective. Since the various parts are clearly treated, I then suggest the paper for publication in its present form.

Author Response

“I have reproduced the results in the paper, which seem correct from my perspective. Since the various parts are clearly treated, I then suggest the paper for publication in its present form.”

Dear Collegue, we are grateful for your high assessment of our work.